# Predicting Consumer Personalities from What They Say

**Hsiu-Yuan Tsao [1], Ching-Chang Lin [2,\*], Hui-Yi Lo [1]** and **Ruei-Shan Lu [3]**

[1] Department of Marketing, National Chung Hsing University, Taichung City 402, Taiwan; jodytsao@dragon.nchu.edu.tw (H.-Y.T.)

[2] Department of Business Administration, Taipei City University of Science and Technology, Taipei City 112, Taiwan

[3] Department of Management Information System, Takming University of Science and Technology, Taipei City 114, Taiwan

**\*** Correspondence: cclin@ba.tpcu.edu.tw

**Abstract:** This study mapped personality based on the newly proposed extraction method from consumers' textual data and revealed the relevance (attention) and polarity (affection) of words associated with a specific personality trait. Furthermore, we illustrate how unique words are used to predict a consumer's behavior associated with certain personality traits. In this study, we employed the scales of the Kaggle MBTI Personality dataset to examine the methodology's effectiveness, extract the personality traits from the textual data into features, and map them into the traits/dimensions of the existing scale. Based on the results obtained in this study, we assert that using the TF-IDF algorithm is a good way to generate a custom dictionary. Furthermore, sentiment scoring with an AI-empowered machine learning algorithm provides useful data to filter and validate more coherent words to understand and, thus, communicate a particular aspect of personality. Finally, we proposed that four situations involving the interaction between attention (frequency) and affection (sentiment) allow us to better understand the consumer and how to use the feature words in terms of the interaction between attention (TF-IDF score) and affection (sentiment score).

**Keywords:** personality traits; sentiment analysis; text analytics; machine learning; MBTI





## 1. Introduction

An awareness of the personalities of those we interact with is beneficial because psychographic segmentation can increase the effectiveness of advertising, promotion, and other marketing activities and improve the measurement of job performance and related functions [1]. Nevertheless, most people assess customers' personality traits using psychological tests. The most widely used measures are the Big 5 model [2] and the MBTI model [3], the latter being a time- and energy-consuming method. Unfortunately, consumers can be reluctant to fill out tedious surveys and, instead, use social media, blogs, or comment threads to post text related to their interests, hobbies, lifestyles, and opinions.

Psychological research suggests that certain personality traits can correlate with linguistic behavior [4]. Furthermore, the automatic detection of personality traits from written messages has attracted significant attention from computational linguists and natural language processors [5]. Term frequency–inverse document frequency (TF-IDF) is a weighting scheme intended to measure how important a word is to a specific document (in our case, a user review) within a collection (or corpus) of documents. This scheme is widely used for information retrieval and summarization. TF-IDF can determine a word's importance by weighing its frequency within a particular document [6]. The highest-scoring words in a document are the most relevant to that document, more so than any other document [7]. Therefore, any personality trait can be regarded as a document. When all the personality traits undergo TF-IDF vectorization, those scores can then be used to classify a user's personality by having each document manually labeled with the aid of a psychology expert

or via a psychological test (such as the well-known MBTI). Therefore, TF-IDF, along with supervised data (via an expert or a psychological test), can provide a score for clusters of words (feature words) that are highly associated with a personality trait.

Past research has been devoted to automatic personality detection via TF-IDF and a machine learning algorithm. However, developing the predictive power of machine learning models that use the same features to predict consumer personality via textual data requires more exploration [1]. In other words, the stream of research devoted to the automatic detection of personality has focused on improving the efficiency and accuracy of personality prediction. However, the intrinsic concept of those features has not been sufficiently explored, and little is known about words that individuals with specific personalities use. Thus, one of the crucial research questions in this study is how to further extract and validate words that might reflect coherent aspects of personality.

Topic modeling, such as LDA, is a popular method for automatically categorizing words reflecting specific personality traits to explore which words a specific personality commonly uses. However, unsupervised topic models, e.g., LDA, often generate incoherent aspects [8]. Furthermore, these existing methods extract many aspects that are not relevant to the domain of interest. Scale-directed text analysis (SDTA) is a new method for generating custom dictionaries for any construct. It can even generate more valid words from constructs; however, the method relies heavily on knowledgeable oversight in the building process [9]. Therefore, it is worthwhile to explore more automatic semi-supervised approaches to develop sound techniques for automatic word extraction to identify consumers' personalities. In addition, the industry needs a rapid automatic dictionary generation method for each construct as well.

First, this study will attempt to extract words based on the TF-IDF scores to generate a dictionary of customer personality traits, as most past research has created. Second, we argue that the core technique of TF-IDF is to count the frequency of words, which is based on the extent of attention rather than the extent of affection (preference or valence) [10]. Hence, this study will consider words associated with a specific aspect and apply sentiment analysis to examine sentences that include words necessary to obtain the affection of an aspect rather than adopting TF-IDF scores to predict or compare the questionnaire ratings [8,11,12]. Previous research has utilized TF-IDF scores alone to compare or predict the questionnaire ratings or for manual labeling, which is not equivalent [9,13]. The main reason for this is that, besides focusing on attention, we also considered affection to be equivalent, comparing it to the results of a questionnaire or psychological test.

Second, the study will employ a sentiment score of featured words instead of only a TF-IDF score to predict the questionnaire ratings. Hence, the other crucial research question is the following. For those sentences of feature words relevant to personality traits, their sentiment score could be an effective source of information to filter and validate more coherent words to understand and, thus, communicate a particular aspect of personality. In other words, we need to identify feature words and the sentiment of the word associated with a personality trait.

Finally, we adopt the strategic analysis grid of FTTA (From Text to Action), which is an analysis framework based on an aspect to discover four interactions of attention (frequency) and affection (sentiment) [10] to further explore how consumers with specific personality traits use those featured words in term of the interaction of attention (TF-IDF score) and affection (sentiment score). The final research question in this study is whether people with specific personality traits intensively use specific feature words positively or negatively.

## 2. Research Methodology

As we mentioned in the Introduction, the personalities of those we interact with are beneficial because psychographic segmentation can increase the effectiveness of advertising, promotion, and other marketing activities [1]. Past research suggests that by taking advantage of insights into psychological factors, marketers can more effectively attract buyers through emotional involvement at the expense of functionality [14]. Additionally,

the consumer-perceived price also varies depending on the psychological traits of each individual [15]. As for automatic personality detection via algorithm and AI technology, some research provided evidence on improving advertising defectiveness [16,17]. Therefore, we attempted to adopt the scales of the Kaggle MBTI Personality dataset to examine the methodology's effectiveness, extract the personality traits from the textual data into features, and map them into the traits/dimensions of the existing scale to better understand what kinds of words are more intensively used for consumers with specific personality traits. The results should be useful for one-to-one advertising message communication.

### 2.1. How The Outcome Variable (MBTI) Is Transformed and Used

First, we employed the well-known scales of the Kaggle MBTI Personality dataset to examine the methodology's effectiveness at extracting the personality traits from the textual data into features and mapping them onto the traits/dimensions of the existing scale. This dataset contained over 8600 rows of data. Each row listed a person's type (the person's four-letter MBTI code/type) and the last 50 items they posted. The data were collected through the Personality Cafe forum (https://www.personalitycafe.com/, accessed on 15 December 2022). A sample of the dataset is shown in Table 1.

**Table 1.** Sample data of Personality Cafe forum dataset.

| Type | Post |
|------|------|
| INFJ | http://www.youtube.com/watch?v=qsXHcwe3krw \|\|\| http://41.media.tumblr.com/tumblr_lfouy03PMA1qa1rooo1_500.jpg \|\|\|enfp and intj moments https://www.youtube.com/watch?v=iz7lE1g4XM4 |
| ENTP | I'm finding the lack of me in these posts very alarming. \|\|\|Sex can be boring if it's in the same position often. For example, me and my girlfriend are currently in an environment where we have to creatively use cowgirl and missionary. There isn't enough... \|\|\|Giving new meaning to 'Game' theory. \|\|\| |
| INTP | Good one _____ https://www.youtube.com/watch?v=fHiGbolFFGw \|\|\|Of course, to which I say I know; that's my blessing and my curse. \|\|\|Does being absolutely positive that you and your best friend could be an amazing couple count? If so, than yes. Or it's more I could be madly in love in case I reconciled my feelings. |

The Personality Cafe forum provides a large selection of people and their MBTI personality types, as well as what they have written. The dataset originated from the Personality Cafe forum in 2017, and its posts are predominantly in English, with an approximate corpus of 11.2 million words in more than 420,000 labelled points. Each row represents the last 50 posts of each user. Several studies exploring the MBTI personality adopted the Personality Cafe dataset to examine textual messages and personality traits. Most of the results indicated that using a dataset with an expert labelling the personality traits seems to be effective. Hence, we decided to utilize the dataset for this study.

The Myers–Briggs Type Indicator (MBTI) is a personality indicator that was developed based on Carl Jung's model. The MBTI assesses 16 different personality types (INTJ, INTP, ENTJ, ENTP, INFJ, INFP, ENFJ, ENFP, ISTJ, ISFJ, ESTJ, ESFJ, ISTP, ISFP, ESTP, and ESFP). They all differ in their characteristics and must be treated differently [3]. Each personality type (listed in Table 2 below) reflects a unique human psychological archetype.

**Table 2.** The definition of dimension and construct for MBTI personality traits.

| Dimension | Construct | Definition |
|-----------|-----------|------------|
| Mind | Introvert (I) or Extrovert (E) | shows how an individual interacts with others. |
| Information | Intuition (N) or Sensing (S) | shows how an individual sees the world and processes information. |
| Decision | Thinking (T) or Feeling (F) | shows how an individual makes decisions and copes with their emotions. |
| Structure | Judging (J) or Perceiving (P) | reflects an individual's approach to work, making decisions, and planning |

Therefore, we collected raw data that could be arranged and scored to look like the figure below.

No. is the sequence number of the subjects; Type is the category of the MBTI personality; and mind, information, decision, and structure are the dimensions of the MBTI personality. The value of 0 for mind indicates the trait of an introvert, and 1 indicates the trait of an extrovert. Please refer to Tables 3 and 4 for the operational definitions of the other dimensions, constructs, and sample data.

**Table 3.** The sample data of the category of four dimensions of MBTI personality traits.

| No | Type | Post | Mind | Information | Decision | Structure |
|---|---|---|---|---|---|---|
| 1 | ENTJ | I was referring to in every careers always a good memory is required, but | 1 | 0 | 1 | 1 |
| 2 | INTJ | I be well, but I feel different psychically and I like it. I'm sure some of | 0 | 0 | 1 | 1 |
| 3 | INTJ | Hell is other people INTJs are often portrayed as villains due to a lack of | 0 | 0 | 1 | 1 |

**Table 4.** The operational definition of four dimensions of MBTI personality traits.

| Dimension | Construct | Indicator |
|---|---|---|
| Mind | Introvert (I) | Value of mind = 0 |
| | Extrovert (E) | Value of mind = 1 |
| Information | Intuition (N) | Value of information = 0 |
| | Sensing (S) | Value of information = 1 |
| Decision | Thinking (T) | Value of decision = 0 |
| | or Feeling (F) | Value of decision = 1 |
| Structure | Judging (J) | Value of structure = 0 |
| | Perceiving (P) | Value of structure = 1 |

*2.2. Generation of a Custom Dictionary for the Construct*

In this study, we attempted to extract feature words based on TF-IDF scores to generate a customer dictionary of construct/traits, as has been carried out in previous research. TF-IDF (term frequency–inverse document frequency) is a statistical measure that evaluates how relevant a word is to a document in a collection of documents [7]. This evaluation is performed by multiplying two metrics: (1) how many times a word appears in a document and (2) the inverse document frequency across a set of documents. The higher the score, the more relevant that word will be in that particular document but not in other documents.

Thus, the obtained MBTI scores indicated the positive or negative dimension of MTBI regarding the mind, information, decision, and structure. We classified those written texts as mind (1), mind (0), information (1), information (0), decision (1), decision (0), structure (1), and structure (0), respectively, and then calculated each word of TF-IDF. We filtered the higher and expected numbers of words to obtain more than 100 words for each construct (please refer to Table 5). Subsequently, we obtained the sample feature words for each dimension, as Table 6 shows (The programming language R provided the package superml (https://www.rdocumentation.org/packages/superml/versions/0.5.5, accessed on 20 December 2022) to easily obtain the score the TF-IDF).

**Table 5.** The sample value of TF-IDF feature words of MBTI personality traits.

| Mind | | | Information | | | Decision | | | Structure | | |
|---|---|---|---|---|---|---|---|---|---|---|---|
| Construct | Word | TF-IDF | Construct | Word | TF-IDF | Construct | Word | TF-IDF | Construct | Word | TF-IDF |
| mind(0) | contains | $2.29 \times 10^{-5}$ | information(0) | trump | $4.40 \times 10^{-5}$ | decision(0) | rhubarb | $1.79 \times 10^{-5}$ | structure(0) | believe | $1.68 \times 10^{-5}$ |
| mind(0) | dirt | $1.98 \times 10^{-5}$ | information(0) | hyper | $3.02 \times 10^{-5}$ | decision(0) | sm | $1.49 \times 10^{-5}$ | structure(0) | fap | $1.28 \times 10^{-5}$ |
| mind(0) | rigid | $1.90 \times 10^{-5}$ | information(0) | rings | $2.83 \times 10^{-5}$ | decision(0) | empathy | $1.20 \times 10^{-5}$ | structure(0) | stoned | $1.28 \times 10^{-5}$ |
| mind(0) | accomplishing | $1.60 \times 10^{-5}$ | information(0) | carried | $2.58 \times 10^{-5}$ | decision(0) | sighs | $1.10 \times 10^{-5}$ | structure(0) | lowered | $1.08 \times 10^{-5}$ |
| mind(0) | composition | $1.60 \times 10^{-5}$ | information(0) | collective | $2.45 \times 10^{-5}$ | decision(0) | snuggles | $1.10 \times 10^{-5}$ | structure(0) | luna | $1.08 \times 10^{-5}$ |
| mind(0) | socialism | $1.52 \times 10^{-5}$ | information(0) | hopeful | $2.39 \times 10^{-5}$ | decision(0) | alienated | $9.96 \times 10^{-6}$ | structure(0) | breasts | $9.86 \times 10^{-6}$ |
| mind(0) | buildings | $1.45 \times 10^{-5}$ | information(0) | atheism | $2.33 \times 10^{-5}$ | decision(0) | cheering | $9.96 \times 10^{-6}$ | structure(0) | devil | $9.86 \times 10^{-6}$ |

**Table 6.** The sample feature words of each dimension of MBTI personality traits.

| Mind(0) | Mind(1) | Information(0) | Information(1) | Decision(0) | Decision(1) | Structure(0) | Structure(1) |
|---|---|---|---|---|---|---|---|
| meditate | banned | destructive | jetplane | bob_toeback | radiation | algebra | energizes |
| dirt | cheaters | hyper | chow | sm | advantageous | fap | find |
| rigid | vous | rings | permissive | empath | devout | stoned | plethora |
| mew | type | heal | barbecued | probs | raping | memy | pufferfish |
| bees | asounds | produce | bitchiest | war | venue | shrooms | query |

*2.3. Categorization and Sentiment Analysis of Textual Data*

This study proactively proposed that the core TF-IDF technique involves counting the occurrence/frequency of words. However, the frequency indicates the extent of attention instead of the extent of affection (preference or valence) [7]. Hence, this study applied the sentiment analysis of those sentences to obtain the affection of the aspect instead of adopting the TF-IDF score to predict or compare the questionnaire ratings [10]. Past research adopted the TF-IDF score or manual labeling to compare or predict the questionnaire ratings, which is different. Except for the attention aspects, we considered affection equivalent to comparing it with the results of a questionnaire or psychological tests [9,18].

The methods of personality measurement regarding sentiment are summarized as follows. For detailed theories and verification methods, please refer to the scale-directed text analysis (SDTA) developed by scholars [9,10]. R and PHP languages are used to develop programs to convert qualitative text content analysis into quantitative marketing scale scores based on the existing marketing scale (This study uses two word datasets, respectively, the AFINN sentiment lexicon (http://www2.imm.dtu.dk/pubdb/pubs/6010-full.html, accessed on 10 November 2022) and MBTI personality as developed by filtering the higher score of TF-IDF as described in a previous paragraph.).

*2.4. Sentiment Analysis of MBTI Personality*

The AFINN sentiment lexicon was used to distinguish the word polarity (positive, negative, or neutral) and the MBTI lexicon was used to distinguish the degree (via an interval scale from −5 to +5) (please refer to Table 7).

**Table 7.** Sample sentences illustrating how to calculate the sentiment score.

| ID | Dimension | Featured Word | Featured Sentence | Emotional Word | Sentiment Score |
|---|---|---|---|---|---|
| 1 | Decision(1) | technologies | specific technologies meant countries denial | denial | −2 |
| 2 | Information(0) | slang | worrying essay read biography Blank pieces paper Scattered stare blink Squirm time minutes slang depends word circumstance altruistic | worrying | −3 |

Sentiment analysis of textual data measures someone's words to determine their feelings. In some cases, it is considered more revealing than surveys because it is a more organic analytical method [8]. The performance of such sentiment classifiers depends on the domain or topic being analyzed [12]. We developed an automatic textual analysis system in the programming languages R and PHP to scan the collected textual data and compared it to the custom dictionaries of MBTI personalities that we developed using TF-IDF statistics. Based on keywords in the dictionaries, the program identifies relevant sentences and assigns each sentence to a construct of the MBTI personality dimension. The textual data for each construct's sentence was analyzed using sentiment analysis of the publicly available AFINN Sentiment Word List. This is a well-known list of English words manually developed by Finn Årup Nielsen, a researcher at the University of Denmark [19]. Specifically, the AFINN word list was used to rate the valence of each sentence using an integer ranging from –5 to +5 based on word strength. Our automated system also identifies and reverses the sentiment scores of sentences containing negative modifiers. Please refer to Table 7 for two examples of categorizing the sentences and scoring the sentiment polarity of textual data.

For example, the sentences shown in Table 7 were written by a participant. The keywords 'technologies' and 'slang' in those sentences can be found in the 'Decision(1)' and 'Information(0)' dimensions of the MBTI personality, respectively. Furthermore, emotional words, in this case, 'denial'-2 and 'worrying'-3, in those sentences were rated by the AFINN.

The categorization and sentiment analysis of the textual data revealed the sentiment score for each document, as shown in the columns mind(0), mind(1), information(0), information(1), decision(0), decision(1), structure(0), and structure(1) in Table 8. Those scores indicate the extent of the valance of the personality traits.

Feature selection is the process of reducing the number of input variables when developing a predictive model. Reducing the number of input variables is desirable to decrease the computational cost of modeling and, in some cases, improve the model's performance. From the perspective of text analytics, feature selection refers to feature word extraction when using the machine learning approach.

Statistical feature selection methods involve evaluating the relationship between each input variable and the target variable using statistics and selecting the input variables that have the strongest relationship with the target variable. These methods can be fast and also effective, although the choice of statistical measures depends on the data type of both the input and output variables. This current study, employing the machine learning approach, uses TF-IDF and sentiment analysis.

However, what is the central criterion to determine the baseline or cut-off threshold to filter more relative feature words for a specific trait? From the perspective of machine learning, feature importance refers to techniques that assign a score to input features based on how useful they are at predicting a target variable. Feature importance scores play an important role in a predictive modeling project, including providing insight into the data, the model, and the basis for dimensionality reduction and feature selection, which can improve the efficiency and effectiveness of a predictive model. Thus, in this study, we attempted to adopt a machine learning algorithm method, Random Forest, to calculate the relative importance of feature words and provide a mechanism to tune the amount and selection features of the words extracted from the TF-IDF and sentiment analysis.

Please refer to Table 9 for the list of feature words and scores of relative importance.

**Table 8.** The data preparation of sentiment score for prediction of MBTI personality traits.

| No | Type | Clean_Post | Mind | Information | Decision | Structure | Mind(0) | Mind(1) | Information(0) | Information(1) | Decision(0) | Decision(1) | Structure(0) | Structure(1) |
|---|---|---|---|---|---|---|---|---|---|---|---|---|---|---|
| 0 | INFJ | http://www.youtube.com/watch?v=qsXHcwe3krw | 0 | 0 | 0 | 1 | 0 | 0 | −15 | 0 | 0 | −15 | 0 | 0 |
| 1 | ENTP | I'm finding the lack of me in these posts very alarming. | 1 | 0 | 1 | 0 | −3 | 0 | 2 | 0 | 0 | −3 | 0 | 0 |
| 2 | INTP | Good one _____ https://www.youtube.com/watch?v=fHiGbolFFGw Of course | 0 | 0 | 1 | 0 | 0 | 0 | 4 | 0 | 0 | 0 | 0 | 0 |
| 3 | INTJ | Dear INTP, I enjoyed our conversation the other day. | 0 | 0 | 1 | 1 | −4 | 0 | −2 | 0 | −2 | 0 | 0 | 0 |
| 4 | ENTJ | You're fired. That's another silly misconception. That approaching is logically is going | 1 | 0 | 1 | 1 | 0 | −2 | 0 | −4 | 1.5 | 0 | 0 | 0 |

**Table 9.** For the example of list of feature words and scores of relative importance.

| Mind | | Information | | Decision | | Structure | |
|---|---|---|---|---|---|---|---|
| **Words** | **Score** | **Words** | **Score** | **Words** | **Score** | **Words** | **Score** |
| hate | 0.004983 | word | 0.00083 | love | 0.039867 | infp | 0.023256 |
| able | 0.003322 | believe | 0.00083 | feel | 0.02907 | makes | 0.01495 |
| made | 0.002492 | talking | 0.00083 | info | 0.020764 | guys | 0.011628 |
| problem | 0.002492 | part | 0.00083 | life | 0.01412 | help | 0.009967 |
| stuff | 0.002492 | start | 0.00083 | feeling | 0.010797 | general | 0.009967 |

*2.5. Validation of a Custom Dictionary for the Construct*

This study employed the score for the sentiment of extracted feature words instead of only the TF-IDF score to predict the questionnaire response. This study adopted the cross-validation function provided by R CARET. Furthermore, XGBoost is an increasingly popular machine learning algorithm due to its high performance and accuracy and its ability to solve overfitting (The programming language R provides easy use and is a powerful CARET package https://cran.r-project.org/web/packages/caret/caret.pdf (accessed on 25 November 2022) to implement the XGBooost algorithm). Before applying the ML algorithm to train and test the data, the input data were prepared as outlined below.

**3. Results**

*3.1. Training Data for the MBTI Personality*

The target variables in this instance were mind, information, decision, and structure, respectively, and the TF-IDF score for the words starting from the column think, know, etc., were the features used to predict the target variable. Please refer to Table 10.

**Table 10.** The TF-IDF score for predicting the MBTI personality traits.

| No | Type | Clean_Post | Mind | Information | Decision | Structure | Think | People | Know | Time | Feel | Love |
|---|---|---|---|---|---|---|---|---|---|---|---|---|
| 0 | INFJ | http://www.youtube.com/watch?v=qsXHcwe3krw | 0 | 0 | 0 | 1 | 0.000 | 0.052 | 0.000 | 0.219 | 0.000 | 0.000 |
| 1 | ENTP | I'm finding the lack of me in these posts very alarming. | 1 | 0 | 1 | 0 | 0.087 | 0.087 | 0.309 | 0.137 | 0.000 | 0.052 |
| 2 | INTP | Good one _____ https://www.youtube.com/watch?v=fHiGbolFFGw Of course | 0 | 0 | 1 | 0 | 0.152 | 0.305 | 0.103 | 0.000 | 0.000 | 0.061 |
| 3 | INTJ | Dear INTP, I enjoyed our conversation the other day. | 0 | 0 | 1 | 1 | 0.137 | 0.137 | 0.174 | 0.072 | 0.000 | 0.000 |
| 4 | ENTJ | You're fired. That's another silly misconception. That approaching is logically is going | 1 | 0 | 1 | 1 | 0.269 | 0.448 | 0.136 | 0.140 | 0.000 | 0.000 |

On the other hand, we prepared a dataset similar to that in Table 11. The target variables were mind, information, decision, and structure, respectively, and the sentiment score for each construct was the features. Please refer to Table 11.

**Table 11.** Data preparation of TF-IDF and sentiment scores predicting the MBTI personality traits.

| Line | Mind | Information | Decision | Structure | Mind(0) | Mind(1) | Information(0) | Information(1) | Decision(0) | Decision(1) | Structure(0) | Structure(1) |
|---|---|---|---|---|---|---|---|---|---|---|---|---|
| 0 | 0 | 0 | 0 | 1 | 0 | 0 | −15 | 0 | 0 | −15 | 0 | 0 |
| 1 | 1 | 0 | 1 | 0 | −3 | 0 | 2 | 0 | 0 | −3 | 0 | 0 |
| 2 | 0 | 0 | 1 | 0 | 0 | 0 | 4 | 0 | 0 | 0 | 0 | 0 |
| 3 | 0 | 0 | 1 | 1 | −4 | 0 | −2 | 0 | −2 | 0 | 0 | 0 |
| 4 | 1 | 0 | 1 | 1 | 0 | −2 | 0 | −4 | 1.5 | 0 | 0 | 0 |

Given the result obtained via two sorts of features, the TF-IDF score and sentiment score of the construct, we can compare the accuracy of the kinds of features possible, as shown in Table 12.

**Table 12.** The comparison of performance prediction between TF-ID and sentiment.

| Metrics | Mind | Information | Decision | Structure |
|---|---|---|---|---|
| TF-IDF | 0.78 | 0.80 | 0.70 | 0.59 |
| Sentiment | 0.80 | 0.80 | 0.72 | 0.61 |
| TF-IDF+Sentiment | 0.80 | 0.91 | 0.74 | 0.66 |

*3.2. Discovering How Consumers Use the Feature Words*

We adopted the strategic analysis grid of FTTA (From Text to Action), which is an aspect-based analysis framework, to discover the four situations of the interaction of attention (frequency) and affection (sentiment) and further explore how a consumer uses those feature words in terms of the interaction of attention (TF-IDF score) and affection (sentiment score).

Tsao et al., 2022 [10], proposed that the data on the topics mentioned in a text (aspect), coupled with the data on the frequency with which they are mentioned (attention) and the sentiment they receive (opinion), can provide useful strategic insights, namely, the FTTA (From Text to Action) grid. This framework is based on a specific aspect or dimension, and the grid explores the interaction between attention and affection based on textual data.

First, the words appearing in the upper right quadrant are characterized by high attention and positive affection, which indicates that those words represent consumers with the corresponding personality traits and more positive affection.

Second, the words in the upper left quadrant, with high attention and negative affection, are most often used by consumers with the corresponding personality traits and negative affection.

Third, the words in the lower right quadrant, with high attention and high affection, indicate highly positive affection, but these are less used by those consumers with a corresponding personality trait.

Fourth, the words in the lower left quadrant, with low attention and log affection, indicate negative affection, and they are also used less frequently.

Please refer to Figure 1 for the sample words in the FTTA grid.

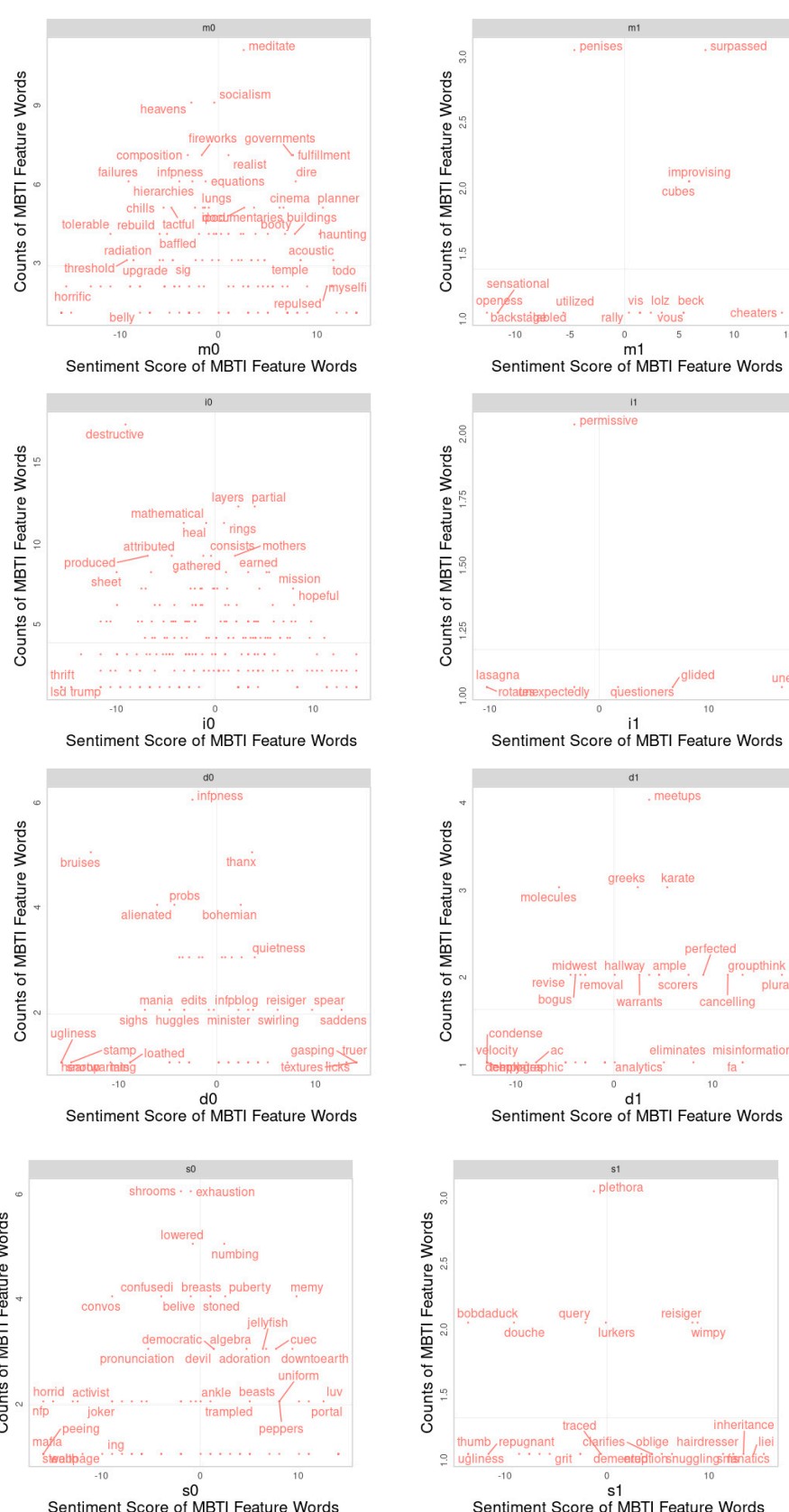

**Figure 1.** FTTA grid of MBTI personality.

We summarized some intensively used words for positive words and negative words for different traits of personality in Table 13.

**Table 13.** Sample words of MBTI Personality.

| | Intensively Used Words | |
| --- | --- | --- |
| **MBTI Dimension** | **Positive Words** | **Negative Words** |
| m0 (I) | realist, fulfilment, tactful, planner, myself | tolerable, haunting, horrific |
| m1 (E) | Improvising, sensational, openness | cheaters |
| i0 (N) | mathematical, produced | heal |
| i1 (S) | questioners, permissive, glided | - |
| d0 (T) | probs, edits | loathed, bruises, saddens |
| d1 (F) | standardized, meetups, devout | misinformation |
| s0 (J) | convos, democratic | query, improves, fanatics |
| s1 (P) | exhaustion, numbing, confused | wimpy, ugliness, lie |

Note. I (Introversion): preferring self-reflection to social interactions and preferring to observe before participating in an activity. E (Extraversion): enjoying socializing and tending to be more enthusiastic, assertive, talkative, and animated. N (Intuition): referring to how people process data. They easily see the big picture rather than the details. S (Sensing): refers to processing data through the five senses. They focus on the present and prefer to "learn by doing" rather than thinking it through. T (Thinking): referring to how people make decisions. They are objective and base their decision on hard logic and facts. F (Feeling): they are more subjective. When making decisions, they consider other people's feelings and take them into account. J (Judging)**:** referring to how people outwardly display themselves when making decisions. They like order and prefer outlined schedules to working extemporaneously. P (Perceiving): they prefer flexibility, live their life with spontaneity, dislike structure, and prefer to adapt to new situations rather than plan for them [20,21].

### 3.3. Summary of Findings

First, in this study, we successfully obtained MBTI scores indicating the positive or negative dimension of MTBI regarding the mind, information, decision, and structure. We could then filter the higher value of TF-IDF for each construct to generate the feature words for each dimension.

Second, given the result obtained via two sorts of features, the TF-IDF score and sentiment score of the construct, we could compare the accuracy of the kinds of features possible via an AI-empowered machine learning algorithm, as shown in Table 12. The results support that the sentiment score is useful for filtering and validating more coherent words to communicate a particular aspect of personality.

Finally, we adopted the FTTA strategy analysis grid, allowing us to better understand the consumer by using the features of words in terms of the interaction between attention (TF-IDF score) and affection (sentiment score). In other words, individuals with specific personality traits tend to heavily use some words positively or negatively, as shown in the upper right and upper left quadrants, respectively, in Figure 1.

### 4. Conclusions

The results obtained in this study confirm that the TF-IDF algorithm can be used to generate a custom dictionary. Furthermore, sentiment scoring with an AI-empowered machine learning algorithm is effective for extracting more coherent words to communicate a particular aspect of personality.

In other words, we attempted to discover the association between words and their sentiments and specific personality traits. The TF-IDF and AI-empowered sentiment analysis can reveal intrinsic concepts of those features and words used by individuals with specific personalities. Furthermore, the strategic analysis grid of From Text to Action (FTTA), which is an analysis framework based on four situations of the interaction of attention (frequency score of TF-IDF) and affection (sentiment), allows us to better understand how consumers use feature words that are positively and negatively associated with personality traits, as Table 13 shows. However, we still require proposing a limitation of the usage of FTTA, that is, how to interpret the feature words is dependent on the realm and context of the research. While a deep dive into the original textual data is required to fully understand

the meaning behind the word mining, a domain expert is also needed to help with the interpretation. However, the FTTA grid still provides a data-driven pathway and cue to lead us to produce the insight. Furthermore, based on the results obtained in this study, a potential further research question could be explored, which is how to achieve automatic awareness of customers' personalities and a one-to-one advertising message-communication strategy [16,17,22].

**Author Contributions:** Conceptualization, H.-Y.T. and C.-C.L.; Methodology, H.-Y.T. and C.-C.L.; Investigation, H.-Y.L.; Resources, H.-Y.L. and R.-S.L.; Data curation, R.-S.L.; Writing—original draft, H.-Y.T.; Writing—review & editing, C.-C.L. All authors have read and agreed to the published version of the manuscript.

**Funding:** This research received no external funding.

**Institutional Review Board Statement:** Not applicable.

**Informed Consent Statement:** Not applicable.

**Data Availability Statement:** https://www.kaggle.com/datasets/datasnaek/mbti-type.

**Conflicts of Interest:** The authors declare no conflict of interest.

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
