# Peer review of "Predicting Consumer Personalities from What They Say"

_applsci, doi:10.3390/app13106148_

Round 1
Reviewer 1 Report
The authors study mapped personality based on the recently proposed method of extraction from consumers’ textual data available in the web. The data processing by machine learning should revealed the relevance (attention) and polarity (affection) of words associated with a specific personality trait. It is a carefully prepared and carried out research work and the authors must be congratulated.
Now, since this work falls in the realm of psychology, it is necessary to extend the literature review and include different systems or spaces for personality mapping. For example, the classification of consumers is frequently based on the Big Five personality domains: The five-factor structure, known as the Big Five (Goldberg, 1981) or the five-factor model (FFM: McCrae and Costa, 1987), emerged and was considered more or less sufficient to encompass the trait-descriptive terms of personality. The most common labels for these domains are extraversion, agreeableness, conscientiousness, negative emotionality (alternatively labeled neuroticism vs. emotional stability), and open-mindedness (alternatively labeled openness to experience, intellect, or imagination); see for example 10.3389/fpsyg.2022.752073
Further, the authors are advised to improve their presentation in a revised version of their manuscript, to make their approach and methodology more understandable by addressing the following issues:
Section 2.1 Please add a paragraph with a description of the PersonalityCafe forum and why you selected this sample for your investigations. Also, explain if all personalities in this forum are classified in the MBTI system and how is the website’s users population divided among the various groups.
Please add more discussion below Figure 1 in order to make it more understandable.
Please improve your literature survey, because this is a popular subject.

Reviewer 2 Report
Thank you for the opportunity to review your manuscript. I appreciate your efforts in the work put into this extensive research Predicting Consumer Personalities from What They Say:
- The information is easy to navigate, and the structure of the paper allows readers to analyze the concepts approached, providing an interesting insight of the topic.
- The paper is written according to academic standards, using proper language and scientific style.
Although, to enhance the quality of the study, it would be wise to pay attention to several issues:
- The introduction should clearly state the research question(s) or the research objective(s) for this paper.
- Authors must include a Literature Review section to provide a background of the topic for the readers. Here, some relevant papers that investigated the subject could be included, for example:
§ Dobrin, C. O., & Girneata, A. (2015). Complaining Behaviour and Consumer Safety: Research on Romania Online Shopping. Economic Studies, 24(1), 161-175.
§ Deac, V., Dobrin, C., & Girneata, A. (2016). Customer Perceived Value-An Essential Element in Sales Management. Business Excellence and Management, 6(1), 43-55.
- I recommend a different section to be included in the paper: “Findings”, where authors can present and detail the findings and importance of the study results.
- The “Conclusions” part must be developed more in order to summarize the main findings of the research, how the research objectives are met through this study and to whom are the results addressed.
- The reference list is not formatted according to the MDPI guidelines. Authors should correct this aspect.
Good luck with your revision!
Reviewer 3 Report
In this paper, the authors proposed methodologies to extract the relevance (by using TF-IDF) and polarity (by SDTA) features from textual data, and use them to predict MBTI personalities.
Major Weakness
1. Lack of technical details:
· How training and testing data is split? When generating the dictionary using TF-IDF and selecting features, are they performed on the training data only or the full data set? It seems to me that they are conducted on the full data set, based on how the authors organized the paper. If so, the information of testing data is leaked during the feature engineering and selection process, hence invalidate the results. The dictionary generation and feature selection should be performed using training data only.
· Section 2-2, for TF-IDF scoring and dictionary generation, please elaborate how this was done. My understanding is that corpus contains all samples. What is a document? Is it all samples (rows in the original data set) corresponding to the same construct, such as Introvert? i.e., about half of the samples is regarded as a document?
· When authors created TF-IDF features and sentiment features, how is TF-IDF feature aggregated for a sample, i.e., at person level? Is it by summing up the TF-IDF scores of the featured words for that person? Has the authors considered the length of user’s post. For example, longer posts are likely to have higher scores, has the scores been normalized based on the sentence length?
· Please elaborate how feature selection is performed in section 2-4, for example, are you using Random Forest for feature selection? What are the predictors and outcome variables for this model? What is the threshold to decide how many words to keep? How many words are selected corresponding to the 4 dimensions in the end?
· For model training, did the authors build 4 separate models, one for each dimension? Are the features pooled/shared for the 4 dimensions?
· Did the authors performed any hyper-parameter tuning in feature selection and model training steps?
· Please elaborate how Figure 1 is made, especially how both attention and affection scores are calculated and aggregated at each word. Please cite a paper if the methodology has been published. Further, the authors state the FTTA allows better understanding about how consumer use feature words, however, it’s difficult to read and interpret these plots. Beyond discussing what are the four quadrants in these figures, I suggest the authors demonstrate and explain with some examples, such as how certain words related to a specific MBTI dimension with respect to attention and affection, and why the FTTA illustration is intuitive or counter-intuitive.
2. Scientific soundness:
· Section 2-4, I’m not convinced the AFINN is an optimal approach to get the sentiment score for a couple of reasons: 1) It appears to me that the emotional word is linked to the featured word if they appear in the same sentence (as demonstrated in Table 7), however, should the position between the emotional word and featured word be taken into account as well? The question here is whether or not the emotional word represents the polarity of the personality traits or the polarity of other content in the sentence. 2) It applies a fixed score to the same emotional word, ignoring the context and difference in individuals. Did the authors considered transformer based neural network models or RNN with an attention mechanism to obtain the sentiment scores?
· Did the authors considerer using a neural network to encode the sentence and predict personality end-to-end? What's the advantages of the proposed method? Are the creation of attention and affection features mainly for better model expandability compared to the neural network-based NLP models?
3. Results not clearly presented:
· Table 1, 3, 5, 6, 8, 9, 10, 11 are screenshots, the font size is inconsistent (because they are screenshot) and the tables are difficult to read. Please consider transform and format them into real tables. Please consider shortening the text column of the tables, by truncating the text, or showing some examples, instead of demonstrating the full text.
· Table 6, structure(0) column is missing from this table.
· Figure 1, please change the layout of this Figure from 3x3 to 2x4 or 4x2, so that subfigures corresponding to the same dimension of MBTI can be compared. Finally, please consider rephrase the y-axis, I’m not sure the meaning of “Counts of Dimension Sentences” and why these counts can be negative.
4. Conclusions not supported by the analysis:
· In Abstract: “Furthermore, we illustrate how unique words are used to predict a consumer’s behavior associated with certain personality traits in the context of the traveling decision.” The authors have not performed any analysis in the context of the traveling decision in the paper.
· In Introduction and Section 2-5, the authors argue “If adopting the sentiment score or sentiment with TF-IDF for those sentences associated with featured words improves the performance of prediction and accuracy, as opposed to using the TF-IDF score, we could assert that the TF-IDF algorithm is a good way to generate a custom dictionary.” I doubt about this argument. When model performance improves with sentiment score, in my opinion, it means that sentiment score provides additional information and is predictive, not necessary TF-IDF is a good way to generate a dictionary. This argument can be made if the authors compared TF-IDF with other approaches to create a dictionary.
Other Suggestions and Questions
1. It’s not clear to me what’s the proposed use case for this study in two aspects, Firstly, how do the authors plan to collect costumers’ textual data (from their social media, blogs and comment threads) and how difficult it is to obtain user consent to use their textual data (if no user consent is explicitly obtained, will it be ethical)? Secondly, could you provide some examples or citations about an awareness of customer’s personalities could improve the effectiveness of marketing activities?
2. I suggest rephrase the title of section 2-1. Section 2-1 mainly discuss how the outcome variable (MBTI) is transformed and used in this study, instead of dictionary generation or feature extraction.
3. Table 8, why in some cells, there are half points, such as 1.5, -2.5? In addition, for example row #1, the person has mind=1, why does the sentiment score corresponding to mind(1) equal to zero? Is it because there is no featured word or emotional word in this person’s post?

Round 2
Reviewer 1 Report
The authors have adequately addressed all my comments and remarks.
Author Response
Appreciate Reviewer1's valuable suggestion and contribution to the revision of paper.
After going through the round 1 revision, we are happy with reviewer1's positive response: "he authors have adequately addressed all my comments and remarks."
Reviewer 3 Report
I thank the authors for making efforts to address my comments. However, a couple of my questions and suggestions were ignored, including: (ordered by the suggestions raised in my original review report)
Major Weakness:
1.5 For model training, did the authors build 4 separate models, one for each dimension? Are the features pooled/shared for the 4 dimensions?
1.6 Did the authors performed any hyper-parameter tuning in feature selection and in model training steps?
3.3 Figure 1, please change the layout of this Figure from 3x3 to 2x4 or 4x2, so that subfigures corresponding to the same dimension of MBTI can be compared. Finally, please consider rephrase the y-axis or provide a footnote, I’m not sure the meaning of “Counts of Dimension Sentences” and why these counts can be negative.
My suggestions and questions of the following are not fully addressed by the author’s response. Let me explain and elaborate on my questions:
Major weakness:
1.1 My question is not about whether or not the authors performed a train/test split. I’m clear that the authors split data for model training and evaluation. However, what about dictionary construction and feature selection? Were they also performed on the training data only, OR on the full data set (training + testing)?
1.2 Based on authors response, it’s now clear that each sample is a document. But I’m confused how Table 5 (feature word with TF-IDF score, by each construct) is generated. If each sample is a document, you will have the TF-IDF score for each word per sample, how is the TF-IDF calculated for each word per construct? Since the purpose is to build a dictionary per construct (as shown in Table 5), is it more convenient to treat all samples belonging to a construct as a document and calculate the TF-IDF scores?
1.3 My question is more about how do you aggregate the features from Table 10 into Table 11, for example, the first row, my understanding is that “think” “people” “know” etc. are feature words and you have the TF-IDF scores for these words, but how do you come up with the aggregated score for “mind(1)”, “information (0)”? Is it by summing up the TF-IDF scores of the featured words for this sample? If so, has the authors considered the length of user’s post. For example, longer posts are likely to have higher scores, has the scores been normalized based on the sentence length?
1.4 What is the threshold (feature important score) to determine how many feature words to keep? How did the author choose the threshold? How many words are selected corresponding to the 4 dimensions in the end (as shown in Table 9)?
1.7 Thanks for pointing me to your other paper. After reading it, I am still not sure my questions are answered. Please elaborate how both attention and affection scores are calculated and aggregated at each word. Further, the authors state the FTTA allows better understanding about how consumer use feature words, however, it’s difficult to read and interpret these plots. Beyond discussing what are the four quadrants in these figures, I suggest the authors demonstrate and explain by using some examples, such as how certain words related to a specific MBTI dimension with respect to attention and affection, and why the FTTA illustration is intuitive or counter-intuitive. For example, the word “turquoise” has high attention and affection score for “decision (0)”, what’s the meaning of that, why is it the case?
2 Given the popularity and recent development in the NLP field (RNN, transformer based neural network models, large language models), the authors decide to use more traditional approach for sentiment analysis and classification. It’s worth discussing the pros & cons for the approach you selected. It’s possible that you could build an end-to-end model (sentence encoder + neural network classifier) and get even better model performance without having to generate a custom dictionary and create the attention and affection features, what’s the advantages of your approach compared to the neural network-based NLP models?
4.2 In Introduction and Section 2-5, the argument of “If adopting the sentiment score or sentiment with TF-IDF for those sentences associated with featured words improves the performance of prediction and accuracy, as opposed to using the TF-IDF score, we could assert that the TF-IDF algorithm is a good way to generate a custom dictionary.” Is not revised. I still doubt about this argument. This, in my opinion, is not supported by your analysis. It could be a valid argument if the authors compared TF-IDF with other approaches to create a dictionary and demonstrate a better performance using TF-IDF than other approaches to generate the dictionary. I suggest removing this argument if the authors do not plan to do additional analysis.
Other suggestions and questions:
1. I am not sure I understand the use case authors described. My understanding is that you could obtain data from TripAdvisor posts and get one model for beach lovers and one model for mountain lovers. What conclusion could the authors make from these models when the two model gives similar prediction or very different predictions? I suggest discussing the use case for your study or citing related publications to provide a use case.
3. Table 8, why in some cells, there are half points, such as 1.5, -2.5?
Round 3
Reviewer 3 Report
I thank the authors for making efforts to address my comments. The response helps clarify my questions. But my concerns persist with respect to the following points:
Major Weakness
1.1 Thanks for the clarification. Because the feature engineering and selection are based on the full data set instead of training data only, the information of testing data could be leaked during the feature engineering and selection process, hence invalidate the results. Could the authors justify the decision to user full data set? In my opinion, the dictionary generation and feature selection should be performed using training data only.
1.6 Thanks for providing your script. Based on the code, hyperparameter tuning was not performed in this study. The authors fixed the hyperparameters (nrounds=50, max_depth=6, etc.), instead of tune and compare several combinations of the hyperparameters. I suggest the authors perform hyperparameter tuning and report model performance at optimal hyperparameters.
1.7 Thanks for adding a new table and paragraph. My point is since FTTA analysis is presented in this paper, the authors should better explain the results and highlight any insights derived from the analysis. When it’s not well explained and findings are not clear, it’s less meaningful to present and include the analysis. If the authors want to include FTTA but does not plan to perform additional analysis or provide better interpretation, I suggest clearly outline the limitations, such as the points in your response “how to interpretate the feature words are depending the realm and context of research” and it requires deep dive into the “original textual data to fully understand the meaning behind word mining from”.
2. I agree that there are pros and cons for each method. My suggestion is to discuss and outline the pros and cons for the method the authors chose for the study, given the popularity and recent development in the NLP field (RNN, transformer based neural network models, large language models). In addition, some of the pros and cons the author summarized are not accurate in my opinion, for example the token-based NLP models can also handle out-of-vocabulary words. Model training does not require large amounts of labeled data due to the technique of self-supervised learning, and there are many open-sourced pre-trained models (such as BERT), which usually just require modest amount of labeled data to finetune for down-stream tasks. Implementation is fairly easy as well (code and models are usually open-sourced). The vanishing or exploding gradients of NN models are solvable and not a major problem.
3.3 Thanks for reformatting the figures as suggested. The subfigures all have the same titles, making it difficult to distinguish which dimension each subfigure represents. Please consider change the title by adding the dimension (such as Mind(0)) into the subfigure title. Please also consider removing the legends (they take up much space, but are not clear).
